# Relaxing Effects of Breathing *Pseudotsuga menziesii* and *Lavandula angustifolia* Essential Oils on Psychophysiological Status in Older Adults

**DOI:** 10.3390/ijerph192215251

**Published:** 2022-11-18

**Authors:** Ya-Hui Chung, Shiu-Jen Chen, Ching-Luug Lee, Chun-Wei Wu, Yu-Sen Chang

**Affiliations:** 1Department of Horticulture and Landscape Architecture, National Taiwan University, Taipei 10617, Taiwan; 2College of Nursing and Health, Kang Ning University, Taipei 11485, Taiwan; 3Department of Horticulture, Hungkuo Delin University of Technology, New Taipei 236354, Taiwan

**Keywords:** essential oils, psychophysiological status, emotion, older adults

## Abstract

We evaluated the effects of breathing *Pseudotsuga menziesii* (*P. menziesii*) and *Lavandula angustifolia* (*L. angustifolia*) essential oils (EOs) during a horticultural activity on older adults. A total number of 92 older adult (71.2 ± 7.7 years old) participants were guided through a leaf printing procedure. In the meantime, water vapor and EOs were diffused in an orderly manner. The heart rate variability-related parameters as well as the brain waves were recorded. In addition, we also collected data for the State–Trait Anxiety Inventory-State (STAI-S) questionnaires before and after the whole indoor natural activity program. The physiological parameters including standard deviation of normal to normal intervals, normalized high frequency (nHF), and high alpha wave increased while the normalized low frequency (nLF), the ratio of LF-to-HF power, high beta wave, and gamma wave decreased following the breathing of *P. menziesii* and *L. angustifolia* EOs. These changes indicated a relaxing effect of breathing both EOs during a horticultural activity on older adults. Our results demonstrated a beneficial effect of *P. menziesii* EO which is as good as a well-known relaxant *L. angustifolia* EO. This notion was supported by the results of STAI-S. Here we developed an indoor natural activity program for older adults to promote physical and mental health.

## 1. Introduction

In modern urbanized society, encounters with nature are valuable and stress-relieving experiences for citizens. Nature-contacting activities such as forest recreation and horticultural experiences are believed to be beneficial to health. For example, recent studies reported that exposure to nature for at least 120 min per week is associated with good health and wellbeing [1,2]. Notably, even as short as 3.5 h of forest walking improves the physical and mental health of middle-aged and older adults [3]. Our group also reported the beneficial effects of various horticultural activities on adult populations [4].

Humans have lived in natural environments for most of their evolution. Therefore, human physiological functions are best suited to the natural environment, which is why the natural environment can enhance human psychophysiological health [5]. Park et al. [6] conducted experiments in 24 different forest environments in Japan. Through the analysis of many research data, the results prove that compared with the urban environment, the forest environment can reduce the concentration of cortisol, pulse rate, and blood pressure; increase parasympathetic nerve activity; and decrease sympathetic nerve activity, allowing the human body to feel relaxed on a physiological level. It also promotes positive emotions, enhances comfort, and lifts spirits.

A study by Li et al. [7] in Japan pointed out that after three days and two nights of forest bathing, the activity of human natural killer cells (NK) was enhanced, along with the number of NK cells and anti-cancer proteins in lymphocytes, and this effect persisted for at least seven days after the activity [8]. Further scientific data revealed: various physiological indicators quantified in forest environments (heart rate variability, pulse rate, blood pressure, anti-cancer proteins, natural killer cell activity, adrenaline, and salivary cortisol concentrations, etc.), compared with indicators of urban environments, support that exposure to forests is more beneficial to human psychophysiological health [9].

During the forest walk, the sense of relaxation and happiness is usually accredited to the sense of smell. It is acknowledged that the soothing and resting effects similar to that of bathing in a forest atmosphere are attributed to the breathing and inhalation of essential oils (EOs) derived from plants. Commercially available EOs are complex and highly volatile compounds that are naturally synthesized by plants as secondary metabolites [10] and extracted by water distillation or cold pressing [11,12]. The chemical compositions of EOs vary depending on the extraction techniques, climate variation, growing region, plant condition, and soil composition [11,13,14,15].

The use of EOs could be traced to the Biblical ages as healing agents for the spirit and body [10,13]. The Greeks and Romans adopted the Egyptian method of using EOs in aromatherapy to improve their quality of life. For example, they used steam baths infused with jasmine, ylang-ylang, and lavender oils to stimulate the central nervous system (CNS) for mental relaxation [16,17]. The aroma of EOs enters the human nose, stimulating the olfactory cells and then converting the fragrance information into neuronal signals, which are transmitted to the limbic system of the brain, such as the hippocampus and amygdala. The hippocampus and amygdala play important roles in regulating the endocrine function and the activity of the autonomic nervous system (ANS), including the sympathetic and parasympathetic subdivisions that affect appetite, emotion, memory, and stress responses [10,12,18]. Odors thus influence the activity of the ANS and regulate emotions, cognition, and behavior [19,20].

Physiological indices, including heart rate variability (HRV) and brain wave activities, are measurable factors that reflect the ANS activity and emotional status. HRV assesses the degree of continuous heart rate flexibility and monitors the ANS response [21,22]. HRV has been suggested as an objective assessment of stress and mental health [23,24,25]. Nerve cells in the brain emit small amounts of electric activity as electromagnetic waves of different frequencies when transmitting messages [26]. These brain wave activities could be recorded and presented as electroencephalogram (EEG) and the patterns of these electromagnetic waves are related to emotions [27,28,29,30]. Therefore, EEG recording can also be used to assess emotional status. The effect of EOs on psychophysiological status could thus be evaluated by analyzing these parameters.

*Pseudotsuga menziesii* (*P. menziesii*) EO is extracted from Douglas firs, which are conifer trees originally grown in North America. The antimicrobial activity of *P. menziesii* EO has been reported [31,32]; however, its other function has not yet been examined. We wondered if the soothing and relaxing effect of a forest bath is attributed to the breathing of the aromas of EO derived from conifer trees such as Douglas firs. In the present study, we compared the function of *P. menziesii* EO with *Lavandula angustifolia* (*L. angustifolia*) EO, which is a well-known mental relaxant [24,33,34,35,36,37,38,39] by analyzing the physiological responses of the participants. Ninety-two older adults of both genders were included in this study and the effects of breathing OEs during an indoor horticultural activity were examined by measuring their real-time HRV and brain waves. We also evaluated the anxiety scores of the participants by collecting State–Trait Anxiety Inventory-State (STAI-S) questionnaires before and after the whole session. Our results showed a strong relaxation effect during breathing of both EOs and suggested a health-promoting indoor natural activity program for older adults.

## 2. Materials and Methods

### 2.1. Participants

The study included 11 male and 81 female participants. The average age was 71.2 (SD = 7.7, range = 60–90). Individuals who have an upper respiratory duct infection or a clinical history of epilepsy, asthma, or allergy to EOs were excluded. After fully understanding the purpose and procedures of the study, all participants were asked to sign the study participant consent form voluntarily. This study was approved by the Research Ethics Committee of National Taiwan University (NTU-REC No. 202104HM009).

### 2.2. Essential Oils

*P. menziesii* and *L. angustifolia* EOs as well as the EO diffusers were purchased from dōTERRA International LLC (Pleasant Grove, UT, USA). The diffusers were used to generate water vapor with or without EO in the experimental environment. The concentration of EO in the diffusor was 2.5% (0.5 mL EO in 20 mL water) as suggested [40]. The major chemical components of each EO provided by the producer are listed in Table 1.

### 2.3. Indoor Horticultural Activity

We chose leaf printing as an indoor horticultural activity for the older adult participants in our study. The leaf printing procedure consists of printing the colors of leaf veins or petals onto cotton fabric by pressing a marble ball [4]. A thin plastic sheet was placed between the leaves or flower petals and the marble ball to reduce the exposure of scent while pressing the ball. The participants were instructed and guided to finish their artwork by experienced experimenters in 60 min.

### 2.4. Experiment Procedure

This study was conducted at the Department of Horticulture and Landscape Architecture, National Taiwan University, between 10:00 a.m. and 4:40 p.m., from 4 November 2021 to 4 January 2022. To ensure the quality of the experiment, the number of participants in each group was limited to 16. To reduce the confounding factors, caffeinated or alcoholic beverages were prohibited; metal objects such as necklaces, watches, earrings, and rings were removed during the whole session.

In this study, we aimed to evaluate the effects of breathing EO during an indoor horticultural activity by monitoring the physiological parameters and collecting questionnaires before and after the whole session. The procedure of the experiment contained seven time periods (Figure 1). During the first period (10 min), participants were guided through the experimental environment and completed the informed consent form of the study. During the second period (15 min), participants were asked to wear patch-type sensors. The HRV and brain waves were continually collected as the baseline data. All participants also finished a State–Trait Anxiety Inventory-State (STAI-S) questionnaire during this period as the pre-test emotional status for reference. In the following period (15 min), an indoor horticultural activity, leaf printing, was introduced to the participants. In the meantime, water vapor without EO was diffused into the air. HRV and brain waves were also continually collected until the end of the experiment. During the next period (15 min), water vapor with *P. menziesii* EO was given via the diffuser. The EO could be inhaled by the participants via normal breathing. During the following 15 min period, between the first and second EOs, water vapor without EO was diffused. Finally, *L. angustifolia* EO was given for another 15 min. At the end of the test, the leaf printing activity also finished and the physiological recordings were then ceased and detached. All participants were asked to complete another STAI-S questionnaire which was used to reflect the post-test emotional status.

During the whole experiment, the indoor air was monitored using an AirBoxx air quality meter (KD Engineering, Glen Cove, NY, USA). The CO_2_ concentration, CO concentration, temperature, and relative humidity in the experimental environment were 889.8 ± 337.7 ppm, 0.18 ± 0.4 ppm, 22.6 ± 2.1 °C, and 74.9 ± 6.4%, respectively.

### 2.5. Physiological Recordings and Analyses

Throughout the experimental period, wearable patch-type sensors for HRV and EEG (BeneGear Inc., New Taipei City, Taiwan) were attached to the chest and forehead, respectively. The HRV and EEG data were continuously stored in the sensors during recording and then transferred to a computer for further processing and analysis.

HRV parameters including HR, SDNN (standard deviation of normal to normal intervals), nLF (normalized low frequency), nHF (normalized high frequency), and LF/HF were analyzed to reveal the ANS activity and emotional status. Brain wave activity was recorded and analyzed by frequencies and power. In the present study, features of alpha (7–13 Hz), beta (13–30 Hz), and gamma (30–100 Hz) waves were explored.

### 2.6. Emotional Evaluation

The State–Trait Anxiety Inventory (STAI) questionnaire includes 40 self-assessment questions, 20 for state anxiety (STAI-S) and 20 for trait anxiety (STAI-T), respectively [41]. The STAI-S is used to assess anxiety levels at a specific time and as an indicator in certain special stressful situations. The STAI-T measures the general feeling of an individual’s anxiety proclivity. In the present study, we used an STAI-S, comprising 20 questions, including ten positive and ten negative items. Each item is scored using the four-point Likert scale, 1 to 4 for negative questions and 4 to 1 for positive questions, respectively. The higher scores indicate higher anxiety levels [42,43].

### 2.7. Statistics

IBM SPSS Statistics for Windows, Version 21.0, was used to analyze all data (IBM Corporation, Armonk, NY, USA). Except for the HR, the sex-dependent differences in all physiological parameters and emotional scores were insignificant; we, therefore, pooled data collected from both genders together in each test and question. Data are expressed as the mean ± standard error (Mean ± SE). Non-parameter statistics: the Wilcoxon signed-rank test was used to compare the differences between baseline and treatment (water or EOs). A *p*-value < 0.05 was considered statistically significant.

## 3. Results

### 3.1. Physiological Recordings and Analyses

To evaluate the effects of breathing EOs during an indoor horticultural activity, leaf printing, a wearable HRV sensor was used to continuously record HR-related parameters. During the leaf printing procedure, water vapor, water vapor with *P. menziesii* EO, water vapor, and water vapor with *L. angustifolia* EO were diffused into the air in an orderly manner. The baseline activity was recorded before the beginning of the horticultural activity. The HRs of the participants were not significantly altered during the whole experiment. However, HRV parameters were affected by the breathing of EOs. Compared with the baseline, the value of the standard deviation of normal to normal intervals (SDNN) was not changed while water vapor was diffused during the procedure of leaf printing; notably, SDNN was increased while *P. menziesii* EO and *L. angustifolia* EO were given (Figure 2A). SDNN represents the intensity of autonomic regulation of the sinus node and thus the overall activity of the autonomic nervous system, and is an indicator of physiological resilience to stress [44,45,46,47,48]. We next analyzed the percentages of normalized low frequency (nLF) and high frequency (nHF) that reflect the sympathetic and parasympathetic activity, respectively [22,24,49]. Diffusion of water vapor did not affect nLF and nHF, however, the percentages of nLF decreased while *P. menziesii* EO and *L. angustifolia* EO were diffused (Figure 2B). Correspondingly, the percentages of nHF increased when the participants were breathing both EOs (Figure 2C). The ratio of nLF to nHF (LF/HF) indicates the autonomic activity balance [23,44]. We also observed decreased LF/HF when both EOs were given whereas water vapor alone did not affect LF/HF (Figure 2D). Together, these results demonstrated a physiological condition of decreased sympathetic and increased parasympathetic activities, indicating a relaxing status during breathing of EOs.

A wearable EEG sensor was also used to continuously record brain wave-related parameters. The baseline activity was recorded before the beginning of the horticultural activity. We analyzed the high alpha (10–13 Hz) wave, which is frequently represented as a relaxed but focused brain wave [29,50,51]. Compared with the baseline, the value of the high alpha wave was increased significantly during the exposures of both *P. menziesii* EO and *L. angustifolia* EO. The diffusion of water vapor without EO did not alter the high alpha wave (Figure 3A). Further, we analyzed the high beta (20–30 Hz) wave, which is commonly associated with stressful situations [52,53]. The high beta wave was reduced while breathing both *P. menziesii* EO and *L. angustifolia* EO (Figure 3B), indicating a reduction in stress level. Interestingly, the value of the high-beta wave in the second water vapor section was significantly lower than the baseline level, suggesting a long-lasting effect of *P. menziesii* EO. We also analyzed the feature of the gamma (30–100 Hz) wave, which is negatively associated with the level of relaxation [54,55,56,57,58]. The gamma waves reduced while the participants were breathing EOs (Figure 3C), indicating a more relaxed condition. Together, these EEG data also suggested an EO-dependent relaxing effect and reduction in stress levels.

### 3.2. Emotional Evaluation

The emotional status of the participants was evaluated using STAI-S questionnaires. The first questionnaire was given to the participants before the horticultural activity and diffusion of EOs as the pre-test emotional status. A second questionnaire was given after the completion of the experiment to evaluate the post-test emotional status. Compared with the pre-test scores, the STAI-S scores of the post-test decreased significantly in all questions (Figure 4). In the scoring system, higher scores denote higher anxiety levels [42,43]. Our findings indicated that the anxiety level of the participants was reduced after the horticultural activity and breathing of EOs.

## 4. Discussion

This study established a procedure to analyze physiological parameters including HRV and brain waves and evaluate the relaxing effects of breathing *P. menziesii* and *L. angustifolia* EOs during a horticultural activity on older adults.

Encounters with nature, such as forest bathing, are healthful. People in natural environments exhibit lower sympathetic activity than in urban environments and increased parasympathetic activity [6,9,59,60]. However, residents in modern society have a hectic pace of life, and time is often limited; forest recreation is usually difficult to achieve. Programs that incorporate elements of the forest environment such as natural light, plant-based features, organic textures, sounds, and aromas to integrate nature into human life, are being developed to promote mental and physical health [59,61]. For example, stress-relieving effects have been shown during indoor horticultural activities [4]. Here, we combined indoor horticultural activities with the breathing of *P. menziesii* and *L. angustifolia* EOs as an indoor natural activity program and aimed to create an indoor natural environment where the participants are immersed in natural elements close to forest bathing.

We analyzed the physiological status of the participants including HRV and brain waves during the indoor natural activity. HRV could be used to evaluate the activity of ANS including sympathetic and parasympathetic divisions. SDNN represents the overall activity of the ANS and denotes an indicator of physiological resilience to stress [44,45,46,47,48]; while the percentages of nLF and nHF reflect the sympathetic and parasympathetic activity, respectively [22,24,49]. Brain waves, on the other hand, could be used to evaluate mental status. The power of alpha waves increases when people are relaxed and at rest [29,62,63,64]. While the values of high beta positively correlated with the stress level [27,63,64], the gamma wave is negatively associated with the level of relaxation [54,55,56].

Breathing of EOs has benefits for mental and physiological status [65]. Our physiological results, including increased SDNN, nHF, and high alpha wave activity accompanied by reduced nLF, LF/HF, and high beta wave and gamma wave during the exposure of *L. angustifolia* EO, indicated a relaxation effect. Our findings are in line with previous reports [24,66,67,68,69]. The main component of *L. angustifolia* EO is linalool (36.03%), which has relaxation and anxiolytic functions [70,71]. Further, its cognitive-enhancing effects such as a decrease in arousal level and an improvement in the sustained component of attention has also been proposed [72]. The benefits of breathing *L. angustifolia* EO are evident.

Compared with the findings during breathing *L. angustifolia* EO, similar physiological results were obtained while defusing *P. menziesii* EO, indicating a relaxation effect. To the best of our knowledge, this is the first trial of using physiological approaches to verify the function of inhaling *P. menziesii* EO. The main components of *P. menziesii* EO include β-pinene (21.16%) and α-pinene (14.97%). Pinenes are monoterpenes that have various functions including antimicrobial activity [73] as well as anti-inflammatory, antioxidant, and neuroprotective properties [71]. Further, the anxiolytic effect of pinenes has been proposed [74,75,76,77,78]. Inhaled pinenes may modulate the serotonergic, adrenergic, and dopaminergic systems in the brain as well as the expression of BDNF in the hippocampus that are involved in the pathogenesis of depression and anxiety [71]. Based on the physiological findings, our results demonstrated a relaxation effect of *P. menziesii* EO; the pinenes could be the active component for this function.

The use of the STAI questionnaires could provide more insight into the subjective psychological state of the participants [42,43,54,60,79,80]. We evaluated the anxiety scores of the participants by collecting STAI-S questionnaires before and after the whole program. As for the results of the STAI-S questionnaires, the scores all decreased after the indoor natural activity program, indicating a stress-relieving effect. This is consistent with the previous findings using STAI-S that the aroma of EOs can reduce anxiety in dental patients [37,39] primiparous women during childbirth [81], and university students [82].

In our present study, *P. menziesii* and *L. angustifolia* EOs were diffused during the horticultural activity. The effects of EO were prominent because the changes in HRV-related parameters and brain waves during EO diffusion diminished when water vapor was diffused. Our goal was to compare the HRV and brain waves in different conditions of the same individuals. In this design, the diffused water vapor was used as the control to avoid affecting the accuracy of relaxation benefits due to differences in the innate physical and mental adjustment ability between individuals. The 15 min duration for water and EO was chosen based on the short-term adaptation property of the olfactory system [83,84,85,86].

### 4.1. Limitations

There are limitations in this study. A total number of 92 participants (11 males and 81 females) was included. The results were mostly collected from females (88%). Although the sex-dependent differences in all physiological parameters and emotional scores were insignificant in this study, we still need to recruit more male participants for comparison. In addition, the wearable EEG device was attached to the forehead in this study and so the brain waves were mostly recorded from the prefrontal cortex. The neural activities in other brain areas were not recorded and analyzed.

### 4.2. Future Research

Similar to linalool, a major component of *L. angustifolia* EO, the cognitive-enhancing effects of pinenes, and major components of *P. menziesii* EO were demonstrated [71]. However, these findings were collected from mice [87] and fruit flies [88]. It is worth testing if *P. menziesii* EO improves learning and memory in humans, especially older adults. Since the sense of smell plays a significant role in the biopsychological effects of stress, mood, and work capacity [20,30,38,89,90], diffusion of *P. menziesii* and *L. angustifolia* EOs could also be suitable for people of other ages. We should extend the study to other age groups in the future.

## 5. Conclusions

In this study, we developed a natural element-based program combining indoor horticultural activity and diffusion of *P. menziesii* and *L. angustifolia* EOs and analyzed the psychophysiological status of the participants. This indoor natural activity program has relaxation and stress-relieving effects and is believed to benefit the health of older adults.

## Figures and Tables

**Figure 1 ijerph-19-15251-f001:**
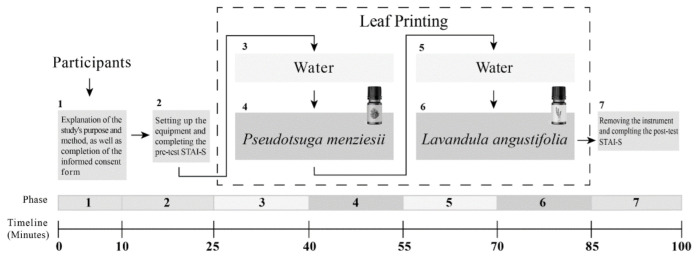
The procedure of the experiment.

**Figure 2 ijerph-19-15251-f002:**
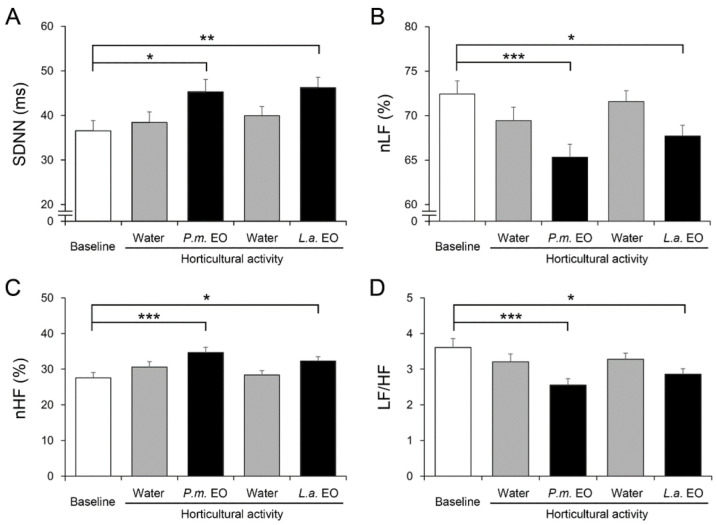
The heart rate variability (HRV)-related physiological parameters. Following the baseline period, an indoor horticultural activity, leaf printing, began. During the leaf printing procedure, water vapor (water), water vapor with *Pseudotsuga menziesii* essential oil (*P.m.* EO), water vapor (water), and water vapor with *Lavandula angustifolia* essential oil (*L.a.* EO) were given in an orderly manner. The values of SDNN (standard deviation of normal to normal intervals) increased during the exposures of both EOs (**A**). The percentages of nLF (normalized low frequency) reduced while the participants were breathing EOs (**B**). Correspondingly, the percentages of nHF (normalized high frequency) increased while breathing EOs (**C**). The LF/HF ratio decreased when both EOs were given (**D**). Data were collected from 92 participants and presented as mean ± SE. *, *p* < 0.05; **, *p* < 0.01. ***, *p* < 0.001, Wilcoxon signed-rank test.

**Figure 3 ijerph-19-15251-f003:**
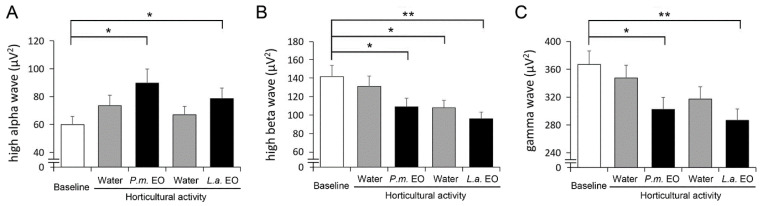
The brain-waves-related physiological parameters. After the baseline period, an indoor horticultural activity, leaf printing, was introduced. In the meantime, water vapor (water), water vapor with *Pseudotsuga menziesii* essential oil (*P.m.* EO), water vapor (water), and water vapor with *Lavandula angustifolia* essential oil (*L.a.* EO) were diffused into the air in an orderly manner. The values of high alpha wave increased during the diffusion of both EOs (**A**). The values of high beta wave decreased while EOs were given (**B**). The gamma wave was reduced while breathing EOs (**C**). Data were collected from 92 participants and presented as mean ± SE. *, *p* < 0.05; **, *p* < 0.01, Wilcoxon signed-rank test.

**Figure 4 ijerph-19-15251-f004:**
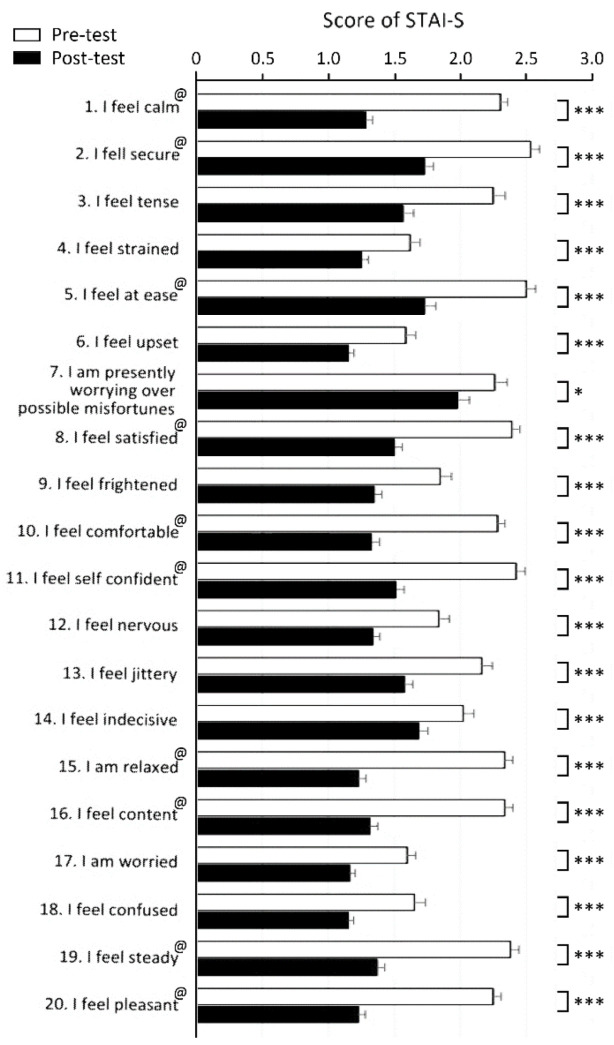
STAI-S scores before and after the horticultural activity and diffusion of EOs. In all questions, the scores decreased dramatically in the post-test STAI-S questionnaire. @ indicates a positive question. Data were collected from 92 participants and presented as mean ± SE. *, *p* < 0.05; ***, *p* < 0.001, Wilcoxon signed-rank test.

**Table 1 ijerph-19-15251-t001:** Major chemical components of *Pseudotsuga menziesii* and *Lavandula angustifolia* essential oils.

*Pseudotsuga menziesii* EO *	*Lavandula angustifolia* EO **
β-pinene 21.16%	linalool 36.03%
α-pinene 14.97%	linalyl acetate 30.67%
terpinolene 11.01%	β-ocimene 4.49%
δ-3-carene 8.87%	lavandulyl acetate 4.33%
sabinene 6.37%	terpinen-4-ol 3.3%
γ-terpinene 4.86%	β-farnesene 2.78%
limonene 3.15%	β-caryophyllene 2.72%
α-terpinene 3.08%	3-octanone 1.32%
myrcene 2.38%	1-Octen-3-yl acetate 1.24%
citronellyl acetate 2.61%	lavandulol 1.01%
β-phellandrene 1.76%	
camphene 1.61%	
α-terpineol 1.41%	
para-Cymene 1.08%	

* Analyzed by gas chromatography–mass spectrometry (GC–MS), a total of 61 chemical components were identified. Only components >1% are listed. ** Analyzed by GC–MS, a total of 51 chemical components were identified. Only components >1% are listed.

## Data Availability

The data will be made available through the request to the corresponding author.

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
