# Peer review of "Relaxing Effects of Breathing Pseudotsuga menziesii and Lavandula angustifolia Essential Oils on Psychophysiological Status in Older Adults"

_ijerph, 2022, doi:10.3390/ijerph192215251_

Round 1
Reviewer 1 Report
General comments
=============
This paper aims to compared the function of P. menziesii EO with Lavandula angustifolia, it provides evidence results showed a strong relaxation effect during breathing of both EOs and suggested a health-promoting indoor natural activity program for older adults. I would like to express several concerns and provide some comments and suggestions as follows. Hopefully, the following comments and suggestions will be helpful for improving this paper.
=============
Major comments
---------------------
1. This paper still lacks further explanation of stress and mental health behavior in relaxing effect of a forest bath, especially the reference to the past research about the environmental Behavior and emotion behavior.
2. For the literature, it is recommended that the literature related to compare studies from tress and mental health behavior in relaxing effect of a forest bath, and to indicate the results of studies in other research.
Minor comments
---------------------
3. This study included 11 male and 81 female participants. Why choose these old samples? Do they have the different with other samples? On the other hand, do the other control variables influence the result? Ex: income or health?
4.The authors can discuss results, implications for managers and scope for future researches in different sections to enhance readability
Author Response
Dear Reviewer:
Please see the attachment.
Best regards,

Reviewer 2 Report
Dear author, I have to congratulate you on a well performed study that evaluated the effects of essential oils as relaxants. The topic is interesting and has several potential uses.
In the study I have encountered several issues that need to be addressed prior to the publishing.
1. The participants group is relatively small and unbalanced as you have already acknowledged in the limitation section. I believe the power calculation would be appropriate in the matter.
2. You have described your statistical methods clearly. I am not convinced the selected methods were appropriate for the sample. You were using t-tests that are suitable for normal distribution. In such a small sample, there is not very likely. The distributions have to be tested and those tests clearly stated or non-parametric tests.
Author Response

(The authors gave the same response as above.)
